# A Phase II Study of Glembatumumab Vedotin for Metastatic Uveal Melanoma

**DOI:** 10.3390/cancers12082270

**Published:** 2020-08-13

**Authors:** Merve Hasanov, Matthew J. Rioth, Kari Kendra, Leonel Hernandez-Aya, Richard W. Joseph, Stephen Williamson, Sunandana Chandra, Keisuke Shirai, Christopher D. Turner, Karl Lewis, Elizabeth Crowley, Jeffrey Moscow, Brett Carter, Sapna Patel

**Affiliations:** 1Department of Melanoma Medical Oncology, Division of Cancer Medicine, The University of Texas MD Anderson Cancer Center, Houston, TX 77030, USA; mhasanov@mdanderson.org; 2Division of Medical Oncology and Division of Biomedical Informatics and Personalized Medicine, Department of Medicine, University of Colorado Anschutz Medical Campus, Aurora, CO 80045, USA; Matthew.rioth@ucdenver.edu (M.J.R.); karl.lewis@ucdenver.edu (K.L.); 3Division of Medical Oncology, Department of Medicine, The Ohio State University Comprehensive Cancer Center, Columbus, OH 43210, USA; Kari.Kendra@osumc.edu; 4Division of Medical Oncology, Department of Medicine, Washington University in St. Louis, St. Louis, MO 63110, USA.; lhernandezaya@wustl.edu; 5Department of Hematology and Oncology, Mayo Clinic Hospital, Florida, Jacksonville, FL 32224, USA; joseph.richard@mayo.edu; 6Division of Medical Oncology, Department of Medicine, University of Kansas Medical Center, Kansas City, KS 66160, USA; SWILLIAM@kumc.edu; 7Division of Hematology and Oncology, Department of Medicine, Northwestern University Feinberg School of Medicine, Chicago, IL 60611, USA; sunandana.chandra@northwestern.edu; 8Division of Hematology and Oncology, Department of Medicine, Norris Cotton Cancer Center, Dartmouth-Hitchcock Medical Center, Lebanon, NH 03766, USA; Keisuke.Shirai@hitchcock.org; 9Blueprint Medicines, Clinical Development, Cambridge, MA 02139, USA; cturner@blueprintmedicines.com; 10Celldex Therapeutics, Inc., Hampton, NJ 08827, USA; ecrowley@celldex.com; 11Cancer Therapy Evaluation Program National Institute of Health (CTEP), Bethesda, MD 20892, USA; jeffrey.moscow@nih.gov; 12Department of Thoracic Imaging, Division of Diagnostic Imaging, The University of Texas MD Anderson Cancer Center, Houston, TX 77030, USA; Bcarter2@mdanderson.org

**Keywords:** uveal melanoma, glembatumumab vedotin, phase II, clinical trial

## Abstract

Glembatumumab vedotin (CDX-011, GV) is a fully human Immunoglobulin G2 monoclonal antibody directed against glycoprotein NMB coupled via a peptide linker to monomethyl auristatin E (MMAE), a potent cytotoxic microtubule inhibitor. This phase II study evaluated the overall response rate and safety of GV, glycoprotein NMB (GPNMB) expression, and survival in patients with metastatic uveal melanoma. Eligible patients with metastatic uveal melanoma who had not previously been treated with chemotherapy received GV 1.9 mg/kg every three weeks. The primary endpoint was the objective response rate (ORR). Secondary endpoints included GPNMB expression, progression-free survival (PFS), overall survival (OS), and toxicity analysis. GPNMB expression was assessed pre- and post-treatment via immunohistochemistry for patients with available tumor tissue. Out of 35 patients who received treatment, two patients had confirmed partial responses (PRs; 6%), and 18 patients had a stable disease (SD; 51%) as the best objective response. 38% of the patients had stable disease >100 days. The grade 3 or 4 toxicities that occurred in two or more patients were neutropenia, rash, hyponatremia, and vomiting. The median progression-free survival was 3.1 months (95% CI: 1.5–5.6), and the median overall survival was 11.9 months (95% CI 9.0–16.9) in the evaluable study population. GV is well-tolerated in metastatic uveal melanoma. The disease control rate was 57% despite a low objective response rate. Exploratory immune correlation studies are underway to provide insight into target saturation, combination strategies, and antigen release.

## 1. Significance

Metastatic uveal melanoma is an orphan disease with no FDA-approved disease-specific treatment options. Immunotherapy and targeted therapy are not as effective as in cutaneous melanoma because of the differences in disease biology and clinical characteristics. This phase II trial of glembatumumab vedotin in patients with metastatic uveal melanoma showed a 57% disease control rate (complete response (CR), partial response (PR), and stable disease (SD)) despite a low objective response rate (6%). Treatment was well-tolerated except for neutropenia, which was managed with granulocyte-macrophage colony-stimulating factor (GM-CSF).

## 2. Background

Uveal melanoma is the most common primary intraocular malignancy in adults and constitutes 3.1% of all melanoma diagnoses. In the United States, the incidence of uveal melanoma has remained stable at 5.1 per million for the past 50 years in contrast to the rising incidence of cutaneous melanoma [1,2]. Though treatment of the primary tumor is highly efficacious, approximately half of the 3000 patients who are diagnosed annually with uveal melanoma will develop metastatic disease [3]. Unfortunately, many of the therapeutic advances that have revolutionized the treatment of cutaneous melanoma have not effectively translated to the uveal melanoma population [4].

The disparity between cutaneous and uveal melanoma is likely attributable to differences in their biology. Whereas cutaneous melanoma is characterized by a UV-mediated high mutation burden and a high incidence of activating mutations in the BRAF protein, uveal melanoma carries a low mutational burden, no UV mutation signature, and a rare occurrence of BRAF mutations. Early canonical activating mutations in the MAPK pathway in *GNAQ* or *GNA11* have not led to the successful targeting of the MAPK pathway [5,6]. Similarly, immunotherapies with checkpoint inhibitors have not demonstrated the same efficacy as in cutaneous melanoma [7]. In previously published work in metastatic uveal melanoma, PD-L1 expression in these tumors was found to be low-to-absent in the majority of cases, which may account for the low response rates to checkpoint blockades [8,9]. This is in contrast to the notable expression of other checkpoint molecules like indoleamine 2, 3-dioxygenase (IDO) and the T cell Ig and ITIM domain (TIGIT) [10]. Regrettably, clinical experience with these latter checkpoint modulators for uveal melanoma is not known. As such, uveal melanoma is left without a standard of care for metastatic disease, and this lack of effective treatment contributes to a median of approximately six-to-thirteen month overall survival rates for patients with metastatic uveal melanoma [11].

Glycoprotein NMB (GPNMB) is a transmembrane protein whose overexpression promotes the invasion and metastasis of cancer cells and is expressed at high levels in uveal melanoma. In preclinical studies, 86% of uveal melanoma specimens demonstrated positive immunohistochemistry (IHC) staining for GPNMB, though with variable intensity [12]. Glembatumumab vedotin (GV) is a fully humanized monoclonal antibody directed against GPNMB that is coupled to the potent cytotoxic microtubule inhibitor monomethyl auristatin E (MMAE) [13]. The proposed mechanism of action is that upon the binding of GPNMB, the complex is internalized, MMAE is released in the lysosome, and tumor cell death occurs as a result of microtubule-inhibition mediated apoptosis. In mouse xenografts with the GPNMB-expressing melanoma cell lines SK-Mel-2 and SK-Mel-5, treatment with GV was found to induce the complete regression of tumors [14,15,16].

Given these promising preclinical results and the biological plausibility of the anti-tumor mechanism, this phase II study was undertaken to determine the effect of single-agent GV on the overall response of patients with metastatic uveal melanoma while also determining the clinical safety, pharmacodynamics, and GPNMB expression of tumors.

## 3. Methods

### 3.1. Study Design

This single-arm, open label phase II trial assessed the efficacy of a single agent, GV, in patients with metastatic uveal melanoma who had not previously been treated with chemotherapy. Enrollment began in January of 2016 and concluded in September of 2017. Seven institutions participated in the trial. All eligible patients were treated with the investigational therapy after obtaining informed consent.

### 3.2. Patient Selection

Patients were considered eligible if they had a histologically or cytologically confirmed metastatic uveal melanoma. Patients with a previous history of uveal melanoma in whom the histologic or cytologic diagnosis was not possible were also allowed at the discretion of the treating investigator. Eligible patients had to be aged greater than 18 years, an Eastern Cooperative Oncology Group (ECOG) performance status of 2 or better, a life expectancy longer than three months, a measurable disease by RECIST version 1.1, adequate organ function, and treatment-naive for chemotherapy. Patients may have received radiation, immunotherapy, and liver-directed or targeted therapy, but all treatment must have been completed 28 days before study drug treatment and prior treatment-related toxicities had to be CTCAEv4 (Common Terminology Criteria for Adverse Events version 4) grade ≤ 1. Both men and women and members of all races and ethnic groups were eligible. The protocol and amendments were approved by the Central Institutional Review Board for the National Cancer Institute under the study ID of 9855. All participants of the study provided written informed consent before initiating study procedures.

### 3.3. Dosing Regimen

All eligible patients were intravenously treated with GV at a dose of 1.9 mg/kg actual body weight every three weeks as outpatients. GV was prepared by dilution in 5% dextrose and administered as a 90-min infusion without premedication. Patients who experienced grade 3 toxicity were treated with reduced doses of 1.3 mg/kg every three weeks and then, if recurrent, at 1 mg/kg every three weeks. An algorithm for the management of hypersensitivity or allergic reactions was provided to investigators.

### 3.4. Response Criteria

Investigator-determined tumor response was radiographically measured every six weeks from treatment initiation using RECIST version 1.1 [17].

### 3.5. Toxicity Capture

Clinical and laboratory assessments were conducted at baseline and every three weeks for up to 30 days following the off-treatment date. Adverse events were graded according to the National Cancer Institute (NCI) Common Terminology Criteria for Adverse Events version 4.0 (https://ctep.cancer.gov/protocolDevelopment/electronic_applications/ctc.htm#ctc_40).

### 3.6. Statistical Analysis

The primary end point was the overall response rate. Secondary end points included overall survival, progression-free survival, clinical safety, tolerability, and pharmacodynamics changes in GPNMB expression by IHC. Patients who received at least one dose of therapy or who experienced objective disease progression during the first cycle of therapy were evaluable for the primary end point and the secondary end points.

A single arm, non-randomized design using Simon’s two stage design was used to evaluate the primary end point. In this design, 18 patients are enrolled in the first stage, and if at least one response is observed in this first stage, up to 14 additional patients are enrolled in the second stage. Assuming 15% of metastatic uveal melanoma samples do not express GPNMB and therefore would not be expected to respond, this study targeted an objective response rate (ORR) of 20% compared to the historical reference ORR of 5%.

Toxicity was reported by type, frequency, and severity according to the NCI Common Toxicity Criteria v4.0. All patients who received any amount of study drug was evaluable for toxicity.

Progression-free survival (PFS) and overall survival (OS) survival curves were calculated using the Kaplan–Meier method. The median PFS and OS were reported with 95% confidence intervals.

### 3.7. Correlative Analysis

Correlative analysis included changes in GPNMB expression via IHC after 1 cycle of GV and a correlation of development of skin rash in cycle 1 with response.

### 3.8. Immunohistochemistry

GPNMB expression was assessed via IHC at Mosaic Laboratories (Lake Forest, CA, USA). The GPNMB (goat polyclonal) IHC assay was designed and validated to be compatible with CLIA guidelines for “homebrew” class I test validation. The procedure for the IHC analysis of GPNMB (goat polyclonal) was performed using automated detection at room temperature (RT) on the Dako Link Autostainer 48. Specimens were sectioned at 3–5-micron thickness, mounted onto positive-charged glass slides, dried, baked, deparaffinized, and rehydrated. Tissue sections then underwent pretreatment using a FLEX Target Retrieval Solution at a high pH (1×, Dako, Catalog# K8004 or S2367, Agilent, Santa Clara, CA, USA) for 40 min in the PT Link (Dako) set to 97 °C. Slides were cooled to 65 °C inside the PT Link, removed, and immediately placed into a FLEX Wash Buffer (Dako, Catalog# K8007) for up to 5 min before placing onto the autostainer. Once the autostainer procedure was initiated, the slides were rinsed with a buffer immediately and after each of the following steps: (1) Incubate with anti-GPNMB antibody or isotype negative control for 30 min; (2) detect with rabbit anti-goat (Vector Laboratories, Burlingame, CA, USA) for 15 min and PowerVision poly AP anti-rabbit IgG for 15 min (rinse for 5 min in a buffer); and (3) stain with a Warp Red buffer solution (BioCare Medical, Concord, CA, USA) for 7 min each. Upon the completion of the staining procedure, slides were counterstained with hematoxylin (Dako) for 2 min followed by a rinse in distilled water, a rinse with a wash buffer for 5 min, and then another rinse in distilled water. Coverslip mounting occurred offline using an automated cover slipper in accordance with Mosaic Laboratories’ standard operating procedures.

Staining was evaluated by a pathologist, and the evaluation of reactivity involved a combination of the following: the cellular localization of staining, staining intensity, subcellular localization, and the percentage of cells staining in the primary component of the tissue type of interest. The GPNMB (goat polyclonal) assay was evaluated on a semi-quantitative scale, and the percentage of cells staining at each of the following four levels was recorded as 0 (unstained), 1+ (weak staining), 2+ (moderate staining), and 3+ (strong staining). The interpretation was performed within the tumor cells of the entire tissue. All tumor cells were considered in the score. A total positive score (percent positive) was derived using variety of magnifications (4×, 10×, and 20×).

## 4. Results

### 4.1. Demographics

Between January 2016 and September 2017, 37 patients were enrolled in the study. The Early Drug Development Opportunity Program (EDDOP) sites were added in August 2016. EDDOP contributed four patients to this effort. Baseline demographics are presented in Table 1. The site of metastases was predominantly the liver, followed by the lung, the lymph node, deep soft tissue, bones, skin, and subcutaneous tissue. The seventh edition M-stage AJCC uveal melanoma – which designates M1a for tumors 3 cm or smaller, M1b for tumors 3–8 cm, and M1c for tumors greater than 8 cm—was used. The enrolled population was balanced for M-stage.

### 4.2. Efficacy

Of 37 patients enrolled in the study, 35 were assigned to treatment with GV. One participant withdrew consent before starting the treatment, and another was ineligible to receive treatment due to elevated liver transaminases. There were two confirmed PRs (6%) via RECIST 1.1 in patients with M1b and M1c disease in the liver, and there were no CRs. An SD was seen in 51% (*n* = 18) as the best overall response in the study, while 40% of the patients experienced the progression of disease (PD) (*n* = 14). One patient did not have a post-baseline tumor assessment after being removed from study due to an adverse event in cycle 1 (grade 4 neutropenia). Another patient without a post-baseline tumor assessment died during cycle 1 due to PD and was counted in the PD rate. The disease control rate—defined as the sum of CR, PR, and SD—was 57% (*n* = 20). Efficacy is presented in Figure 1 as a waterfall plot and in Table 2 as text. The median duration of response was 8.6 months (263 days with a range of 149–377 days). Three patients had new lesions appear at their first tumor assessments, thus qualifying these responses as the progression of disease despite reductions in target lesion(s). An SD lasting longer than 100 days was noted in 34% (*n* = 12) of the total study population, representing the majority of patients with an SD. The median duration of the SD across the study was 4.8 (147 days) months. Figure 2 shows comparative computerized tomography images of metastatic uveal melanoma tumors before and in response to GV treatment.

The median overall survival (Figure 3) for the evaluable population was 11.9 months (95% CI: 9.0–16.9). The median progression-free survival was 3.1 months (95% CI: 1.5–5.6). The range of OS was 0.5–40.2 months, whereas the PFS ranged from 0.5 to 30.5 months.

### 4.3. Adverse Events

All 35 patients who received GV were evaluable for toxicity. Common treatment-related adverse events (TRAEs) are presented as groups by organ class in CTCAE grades (Table 3). The most common TRAEs were skin and subcutaneous AEs, namely alopecia (80%), maculopapular rash (54%), and pruritus (51%), as well as (less frequently) dry skin (9%), skin hypopigmentation (9%), and acneiform rash (6%). Leukopenia (69%), neutropenia (60%), anemia (40%), lymphopenia (17%), and thrombocytopenia (14%) were the most common hematologic AEs; meanwhile, elevated alanine aminotransferase/aspartate aminotransferase (ALT/AST) (63%), nausea (51%), diarrhea (31%), elevated alkaline phosphatase (26%), constipation (23%), oral mucositis (17%), and vomiting (14%) were the most common gastrointestinal TRAEs. Other common TRAEs were fatigue (57%), peripheral neuropathy (43%), anorexia (37%), and arthralgia (26%). The most common grade 3/4 AE was neutropenia (48%), which was reversible and managed with the use of GM-CSF in subsequent cycles. Other grade 3/4 AEs occurring in 1–2 (3–6%) patients were leukopenia, elevated ALT/AST, nausea, diarrhea, constipation, vomiting, rash, fatigue, weight loss, hyponatremia, hypophosphatemia, and hypotension. There was only one grade 5 TRAE reported as encephalopathy, which resulted in death during the first treatment cycle. Upon further review of the data entry, this death was noted as disease progression in the liver with resultant hepatic encephalopathy, and it was not clearly related to GV. Twenty-seven serious adverse events (SAEs) were reported to the serious adverse event portal. Among these SAEs, three (9%) events of neutropenia; two (6%) events of hyponatremia, vomiting, and maculopapular rash; and one (3%) event of elevated AST, diarrhea, hypotension, abdominal pain, nausea, constipation were reported as grade 3 SAEs. The reported grade 4 SAEs comprised seven (20%) cases neutropenia, one (3%) case of leukopenia, and one (3%) case acute kidney injury. There was only one (3%) grade 5 SAE with encephalopathy, as detailed above.

Only one of the responders developed rash during the first cycle of the treatment, but it should be noted that rash occurring in cycle one did not correlate with response or PFS, as demonstrated in previous studies [18,19].

### 4.4. Gpnmb Tissue Expression

Of the 32 patients, 26 (81%) had tissues available for baseline, and 24 (75%) had tissues available for GPNMB expression after one cycle of treatment. In baseline tissue, GPNMB was highly expressed across all metastatic tissues, with 11 (42%) out of 26 available tissues demonstrating expression in 100% of tumor cells, whereas 8 (31%) of 26 tissues exhibited expression in 20–95% of tumor cells. Seven (27%) of the baseline tissues did not demonstrate GPNMB expression. In the evaluation after one cycle of GV (Appendix A), 8 (33%) out of 24 available tissues demonstrated expression in 100% of tumor cells, and two (8%) of them demonstrated zero expression in tumor cells, all unchanged from baseline. While nine (38%) tumor samples showed paradoxical increases in GPNMB expression, three (13%) had a decreased expression from baseline. Two patients (8%) did not have a baseline tissue available to compare.

## 5. Discussion

This phase II trial of GV was conducted to evaluate its safety and efficacy in patients with metastatic uveal melanoma. The best objective response rate was 6%, with no CR. However, there were 56% of patients with an SD, most of whom had an SD > 100 days. The median PFS was 3.1 months, and the median OS was 11.9 months. The side effect profile was tolerable with grade 3–4 neutropenia occurring in 56% managed with GM-CSF.

The objective response rate to GV was not very improved compared to immunotherapy with PD-1 and PD-L1 antibodies (3.6%) [7] or targeted therapy with selumetinib and dacarbazine (3%) [20].

The median PFS in this study was higher than what has been seen in past studies with checkpoint blockade and targeted therapy [5,7,20]. A median OS of 11.9 months was likewise improved over past metastatic trials for this patient population. However, this is still a minimal improvement over historical cohorts and patients who only opt for the best supportive care [21].

The toxicities seen in this study have been previously reported in both cutaneous melanoma and triple-negative breast cancer [18,19]. Unlike those studies, the development of a rash in cycle 1 did not emerge as a biomarker for the response or clinical benefit. Elevated liver transaminases were likely higher in this study due to the common presence of hepatic metastases in uveal melanoma and, therefore, in our study population (91%).

The baseline expression of GPNMB (73%) appeared to be lower than the previous work of Williams et al. that demonstrated an 86% expression of GPNMB in primary uveal melanoma tissues [12]. In breast and cutaneous melanoma, the expression of GPNMB is far lower, and GPNMB expression has been observed to be stable over time [19]. However, we observed a paradoxical increase in expression of GPNMB in 38% of the tumor tissues after one cycle of GV treatment. This raises the possibility of inadequate target saturation by GV at a dose of 1.9 mg/kg in this disease population with a high percentage and intensity of GPNMB staining per tumor. Alternative mechanisms may be at play, thus leading to the upregulation of target.

The results of the study should be assessed within its limitations. First, this study had a small sample size. A current lack of standard treatment modalities for metastatic uveal melanoma makes it challenging to compare treatment efficacy. However, we compared the efficacy of GV to the best available efficacies amongst the clinical trials completed in the past. Though we had a high number of tumor tissue evaluated, the baseline and/or post-treatment tissue samples of some of the patients were not available for GPNMB expression.

## 6. Conclusions

In summary, GV was well-tolerated in the metastatic uveal melanoma patient population. The disease control rate was high and sustained despite a low objective response rate. Combination treatment strategies with immune checkpoint inhibitors are of interest, as are antibody–drug conjugates against other possible uveal melanoma targets.

## Figures and Tables

**Figure 1 cancers-12-02270-f001:**
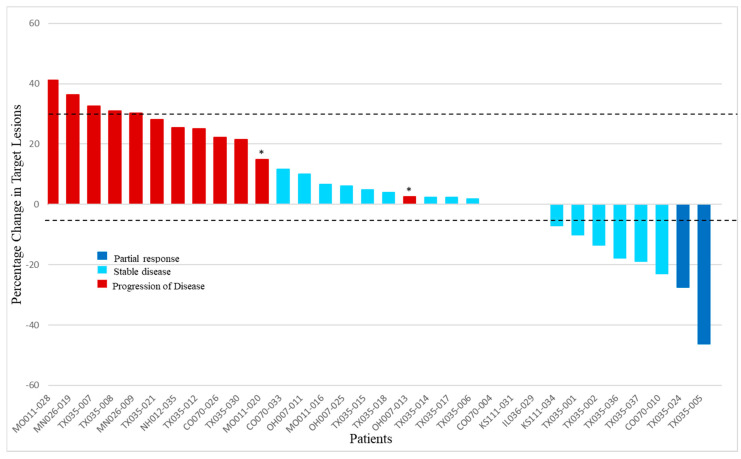
Best overall response of target lesions to glembatumumab vedotin. For each evaluable patient, the best response (defined by shrinkage in RECIST target lesions) is indicated (*n* = 32). Dark blue bars indicate the patients with partial response, light blue columns show the patients who had stable disease, and red columns represent patients with the progression of disease. Asterisks indicate the patients who had appearance of new lesions at the first tumor evaluation. Of the 3 patients not shown, one patient died during cycle 1 due to the clinical progression of disease without post-baseline tumor assessment, and another was removed for grade 4 neutropenia and acute kidney injury during cycle 1 and did not have post-baseline tumor assessment. The third patient had a complete disappearance of target lesion(s) at the first post-baseline tumor assessment but was noted to have new tumors elsewhere, so the best response was classified as the progression of disease. This patient did not have tumor measurements entered into the database and was therefore not included in the graphical output.

**Figure 2 cancers-12-02270-f002:**
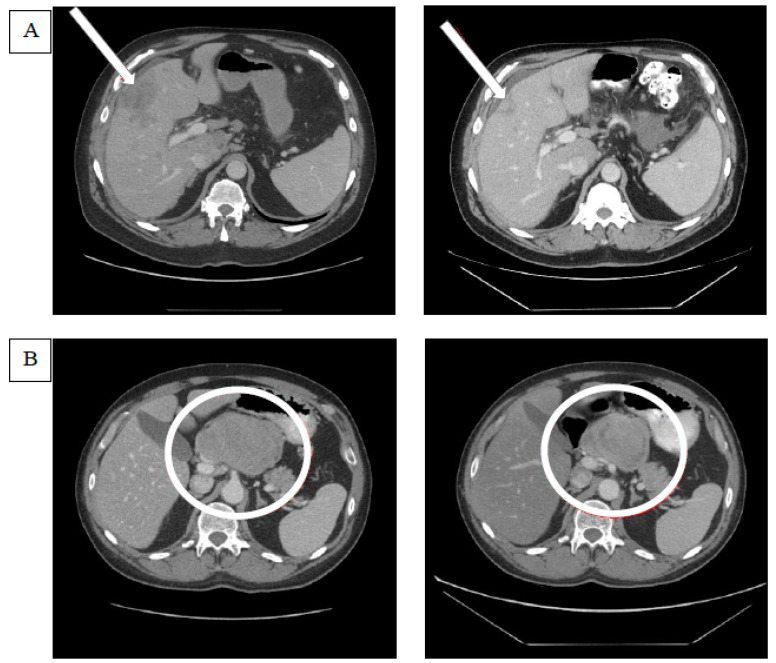
Examples of tumor responses after glembatumumab vedotin treatment. (**A**) The computerized tomography (CT) scan on the left indicates the baseline target lesion in the liver parenchyma that is pointed at by an arrow. The CT scan image on the right demonstrates the shrinkage in the target lesion after glembatumumab vedotin (GV). (**B**) The CT scan on the left demonstrates the target lesion in the porta hepatis lymph node, and the one on the right shows the changes in the lesion after GV. Though the lesion appears to have shrunk in size, the decrease in the lesion’s short axis was not sufficient to meet the response criteria of RECIST 1.1. The target lesion response in the right panel was classified as a stable disease.

**Figure 3 cancers-12-02270-f003:**
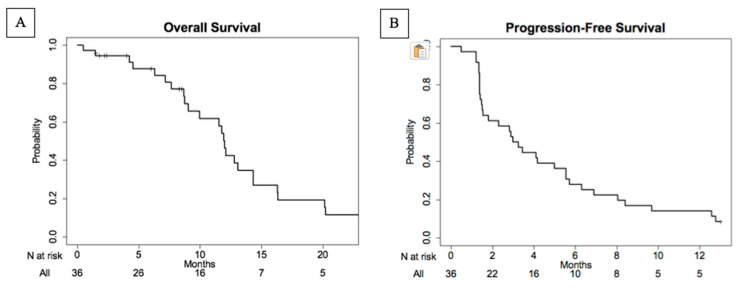
Kaplan–Meier plots of overall survival (**A**) and progression-free survival (**B**) in evaluable patient population. The median overall survival was 11.9 months (95% CI: 9.0–16.9) ranging between 0.5 and 40.2 months. The median progression-free survival (PFS) was 3.1 months (95% CI: 1.5–5.6) and ranged between 0.5 and 30.5 months.

**Table 1 cancers-12-02270-t001:** Baseline patient characteristics.

Characteristics	No. of Patients	%
Age in years		
Median	62	
Range	29–79	
Gender		
Male	18	51
Female	17	49
Performance Status (ECOG)		
0	26	74
1	9	26
M-Stage *		
M1a	16	46
M1b	12	34
M1c	7	20
Sites of Metastasis **		
Liver only	17	48
Extrahepatic only	3	9
Both liver and extra-	15	43
Hepatic		
Extrahepatic sites		
Lung	14	40
Lymph node	9	26
Bones	6	17
Peritoneum/Soft tissue	5	17
Subcutaneous	2	14
Muscle	1 each	6
All other sites ***		3

Abbreviation: ECOG, Eastern Cooperative Oncology Group. * 7th edition AJCC M-stage uveal melanoma: M1a for tumors 3 cm or smaller, M1b for 3–8 cm tumors, and M1c for tumors greater than 8 cm. The enrolled population was balanced for M-stage. ** The most common site of metastasis was the liver. Patients could have more than one site of metastasis. *** Other extrahepatic sites were the brain, the orbit, the thyroid, the heart, the stomach, the spleen, the gallbladder, the adrenal gland, and the kidney.

**Table 2 cancers-12-02270-t002:** Efficacy analyses.

Response	No.	%
Evaluable for response	(*n* = 35)	
Complete Response (CR)	0	0
Partial Response (PR)	2	6
Stable Disease (SD)	18	51
Disease Progression	14	40
Unknown *	1	3
Disease Control Rate (CR + PR + SD)	20	57
Stable Disease > 100 days	12	34
Duration of response, days		
Median	263	
Range	149–377	
Duration of stable disease, days		
Median	147	
Range	82–426	

RECIST 1.1 criteria was used to determine tumor response. * This participant was removed from study after experiencing an adverse event in cycle 1. There was no post-baseline tumor assessment.

**Table 3 cancers-12-02270-t003:** Treatment-related adverse events.

Adverse Event	Any Severity		CTCAE Grade 3		CTCAE Grade 4	
	No.	%	No.	%	No.	%
Hematologic						
Leukopenia	24	69	1	3		
Neutropenia	21	60	12	34	1	3
Anemia	14	40			5	14
Lymphopenia	6	17				
Thrombocytopenia	5	14				
Gastrointestinal						
Elevated ALT/AST	22	63	1	3		
Nausea	18	51	1	3		
Diarrhea	11	31	1	3		
Elevated ALP	9	26				
Constipation	8	23	1	3		
Mucositis, oral	6	17				
Vomiting	5	14	2	6		
Hyperbilirubinemia	4	11				
Abdominal pain	2	6				
Dry mouth	2	6				
Dyspepsia	2	6				
Oral pain	2	6				
Skin and Subcutaneous						
Alopecia	28	80				
Rash, maculopapular	19	54	2	6		
Pruritus	18	51				
Dry skin	3	9				
Skin hypopigmentation	3	9				
Rash, acneiform	2	6				
General						
Fatigue	20	57	1	3		
Pain	6	17				
Chills	2	6				
Flu like symptoms	2	6				
Localized edema	2	6				
Weight loss	2	6	1	3		
Nervous System						
Peripheral Neuropathy	15	43				
Dysgeusia	8	23				
Headache	3	9				
Dizziness	2	6				
Musculoskeletal						
Arthralgia	9	26				
Myalgia	6	17				
Pain in extremity	3	9				
Generalized muscle weakness	2	6				
Respiratory						
Dyspnea	5	14				
Renal and Electrolytes						
Hyponatremia	5	14				
Hypophosphatemia	4	11	2	6		
Hypokalemia	3	9	1	3		
Hyperkalemia	2	6				
Metabolism and Nutrition						
Anorexia	13	37				
Hypoalbuminemia	3	9				
Hyperglycemia	2	6				
Vascular						
Hot flashes	3	9				
Hypotension	3	9	1	3		

Abbreviations: CTCAE: Common Terminology Criteria for Adverse Events, ALT: alanine aminotransferase, AST: aspartate aminotransferase, ALP: alkaline phosphatase. The treatment-related AEs had overall incidences ≥3%, and all had grade 3–4 severity. Only one patient had a grade 5 treatment-related AE with encephalopathy. Empty data represent no reported toxicity.

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
