# Peer review of "A Phase II Study of Glembatumumab Vedotin for Metastatic Uveal Melanoma"

_cancers, 2020, doi:10.3390/cancers12082270_

Round 1

Reviewer 1 Report

The manuscript entitled “A Phase II Study of Glembatumumab Vedotin for 2 Metastatic Uveal Melanoma” reports on a clinical trial with NMB coupled to monomethylauristatin E for the treatment of metastatic uveal melanoma. Survival after diagnosis of metastatic uveal melanoma has not significantly changed in decades, therefore, testing new therapeutical approaches is highly interesting. The present non randomized single arm study has clear limitations for the evaluation of the efficacy of a drug. Comparison to other clinical trials has been performed in order to estimate the efficacy that was, however, rather limited.

Major remarks:

The statement “The protocol and amendments were approved by relevant institutional review boards” is not sufficient. The precise institutional body that granted approval and its ID must be reported.

It is not clear why the authors did not take GPNMB expression as an enrolment criterion since in the absence of expression, no response was expected. This is also concerning ethical issues since GPNMB patients were exposed to drug related toxicity in the absence of a chance of response.

Molecular characterization of the metastases and/or the primary tumors they are derived from has not been reported. The authors should indicate mutational status and chromosome 3 status of the metastases of all patients enrolled and response data should be stratified on the base of molecular data.

Response data should be further stratified by localization of the metastases (liver only, other tissues only, both).

Minor remarks:

Figure 1 should be presented in colors.

State why two patients were not evaluable.

Reviewer 2 Report

Dear Authors,

thank you for this interesting manuscript. It is a privilege to read it. Your research is well planned, well executed and well presented. Unfortunately, the results are somewhat disappointing, which is certainly a situation we are used to when it comes to metastatic uveal melanoma. Besides this, I only have some minor comments, stated point-by-point below.

Best regards

Abstract:

The sentence ”This Phase II study evaluates the overall response rate and safety of GV, glycoprotein NMB (GPNMB) expression, survival analyses in patients with metastatic uveal melanoma.” is a bit strange. The study does not evaluate survival analyses, right? It evaluates survival, period. Please consider rephrasing.

Similarly, the next sentence could be rephrased. No previous chemotherapy was a criterium for eligibility. Perhaps change to “Patients with metastatic uveal melanoma who had not been treated with chemotherapy were included. They received GV…” or similar.

Background:

After the following sentence: “Similarly, immunotherapies with checkpoint inhibitors have not demonstrated the same efficacy as in cutaneous melanoma.” it could be added that this may be a result of low expression of CTLA-4 and PD-1 in uveal melanoma, whereas other checkpoints such as IDO and TIGIT are in fact expressed.

Methods:

You state that the study protocol is “Supplement 1”. I can see “Supplementary table 1” with the GPNMB expression before and after treatment, but no “supplement 1” with the protocol?

Patient selection:

Your protocol states that patients were eligible of they had metastatic uveal melanoma, but in the first sentence of this section you state that they were eligible if they had metastatic uveal melanoma OR locally recurrent uveal melanoma? Does this mean locally recurrent primary uveal melanoma, i.e. without metastases? I’m sure this is not the case, but this needs to be clarified and presented more distinctively.

Correlative analysis:

The word immunohistochemistry on page 4 row 5 seems to be a title but has not been formatted as such?

Immunohistochemistry data analysis

Please elaborate how the cells were scored. How many cells were analyzed? In how many high-power-fields at what magnification? If not all available cells were counted, how where they selected?

Efficacy:

What was the reason that two patients were inevaluable for response?

Adverse events:

Where there no SAEs? It seems not, but please state this clearly with “there were no SAEs” or similar.

Discussion:

The discussion is excellently written

Figure 3:

Please end Kaplan-Meier curve when <10 % of original sample remains (3 patients at risk)

Round 2

Reviewer 1 Report

The authors adequately addressed the issues raised. Still I believe, however, that it should be possible to reproduce figure in colors in the online version which should not determine an increase of publication costs. Please check with the editorial assistant.  
